# Effect of Acute Thermal Stress Exposure on Ecophysiological Traits of the Mediterranean Sponge *Chondrilla nucula*: Implications for Climate Change

**DOI:** 10.3390/biology13010009

**Published:** 2023-12-22

**Authors:** Mar Bosch-Belmar, Martina Milanese, Antonio Sarà, Valeria Mobilia, Gianluca Sarà

**Affiliations:** 1Laboratory of Ecology, Department of Earth and Marine Sciences (DiSTeM), University of Palermo, Viale delle Scienze 16, 90128 Palermo, Italy; valeria.mobilia@unipa.it (V.M.); gianluca.sara@unipa.it (G.S.); 2Studio Associato Gaia, Piazza della Vittoria 15/23, 16121 Genoa, Italy; m.milanese@studioassociatogaia.com (M.M.); a.sara@studioassociatogaia.com (A.S.)

**Keywords:** porifera, Mediterranean, warming, stressor, respiration, clearance

## Abstract

**Simple Summary:**

Climate change is significantly impacting the structure and functioning of marine communities through temperature rise and the repeated occurrence of temperature spikes and marine heat waves. The present study investigated the metabolic responses of the sponge *Chondrilla nucula*, an important component of benthic ecosystems in the Mediterranean Sea, to temperature. Organisms were exposed to six different temperatures ranging from 15 to 32 °C, and respiration and clearance rates were measured. The results revealed temperature-dependent effects on both traits. Higher temperatures correlated with increased respiration rates, peaking at 26 °C. In contrast, clearance rates declined beyond 26 °C, signifying reduced food intake not mirrored by respiration rates. This response at high temperatures implied a potential negative energy balance, suggesting vulnerability to chronic or repeated thermal stress. Consequently, this study predicted the susceptibility of *C. nucula* to climate change conditions, with potential repercussions for its metabolic performance, ecological role, and the associated ecosystem services it provides.

**Abstract:**

As a result of climate change, the Mediterranean Sea has been exposed to an increase in the frequency and intensity of marine heat waves in the last decades, some of which caused mass mortality events of benthic invertebrates, including sponges. Sponges are an important component of benthic ecosystems and can be the dominant group in some rocky shallow-water areas in the Mediterranean Sea. In this study, we exposed the common shallow-water Mediterranean sponge *Chondrilla nucula* (Demospongiae: Chondrillidae) to six different temperatures for 24 h, ranging from temperatures experienced in the field during the year (15, 19, 22, 26, and 28 °C) to above normal temperatures (32 °C) and metabolic traits (respiration and clearance rate) were measured. Both respiration and clearance rates were affected by temperature. Respiration rates increased at higher temperatures but were similar between the 26 and 32 °C treatments. Clearance rates decreased at temperatures >26 °C, indicating a drop in food intake that was not reflected by respiration rates. This decline in feeding, while maintaining high respiration rates, may indicate a negative energy balance that could affect this species under chronic or repeated thermal stress exposure. *C. nucula* will probably be a vulnerable species under climate change conditions, affecting its metabolic performance, ecological functioning and the ecosystem services it provides.

## 1. Introduction

The Mediterranean Sea stands out as a recognized biodiversity hotspot [1] as it hosts a rich biota that includes cold-temperate and subtropical species [2]. Despite it representing a small proportion of the total ocean volume, its distinctive geomorphological history has resulted in an exceptionally high biodiversity level, hosting a significant percentage of all known marine species and a significant proportion of endemic species [3,4]. Moreover, it is anticipated to be among the regions most susceptible to the impacts of climate change (CC) drivers, with a particular emphasis on the pervasive influence of ocean warming [5]. Widely acknowledged as a valuable model, the Mediterranean Sea serves as a crucial arena for assessing the ecological repercussions of climate change on marine biodiversity. It also serves as an experimental platform for exploring potential adaptation and mitigation strategies that hold promise for broader application on a global scale [5].

The frequency of extreme weather events such as droughts and heat waves has doubled worldwide [6], accompanied by an extension in the duration, frequency and severity of their impact, which is expected to keep increasing in the future [7]. The rise in marine heat wave (MHW) occurrences was linked to the swift escalation of mass mortality events on a global scale [8,9]. These dramatic events stood out as one of the most prevalent biological impacts of climate change in the Mediterranean Sea [5,10]; even short-duration temperature anomalies had detrimental effects on marine diversity. In the summers of 1999, 2003, 2008, and 2017, above normal surface water temperatures led to disease outbreaks and mass mortality events that affected numerous benthic species [11]. Reached temperatures ranged between 1 and 3 °C above the mean and maximum climatic values recorded in the studied areas, and the main affected taxa included cnidarians and porifers [8,9,12,13,14,15,16], with up to 80% gorgonian mortality and 90% sponge mortality in some locations [14].

Sponges are key components of benthic habitats worldwide, as they contribute in several ways to ecosystem functioning [17]. Their remarkable ability to process substantial volumes of water and efficiently eliminate both particulate and dissolved organic carbon stands out as a crucial ecological function. Furthermore, sponges make significant contributions to substrate consolidation and function as vital habitats for a diverse array of species [18,19,20,21]. Sponges can be the dominant fauna in shallow tropical and temperate reef habitats, where they can be found in high densities [22,23,24]. Within temperate rocky communities, sponges exhibit formidable competitive capabilities for spatial occupation, outcompeting other invertebrates [25]. Given their indispensable ecological functions, any diminution in the abundance, biomass, and species richness of sponges can potentially trigger cascading impacts, profoundly influencing the structural and functional dynamics of marine ecosystems [17,26].

As the primary driving force behind biological processes, temperature can significantly influence the metabolism, thermal tolerance limits, and distribution patterns of ectotherms, as well as the strength of biotic interaction within their communities [27]. Specifically, environmental warming, driven by temperature changes, may play a pivotal role in this transformative context, given that all biological rates are temperature-dependent [28,29]. Even slight increases in temperature can markedly affect individual metabolic rates and population growth; then, the cascading effects of climate change impacts would ultimately manifest in the functioning of ecosystems and the services they provide, consequently influencing economically significant marine human activities dependent on these ecosystems. As the Earth’s climate undergoes unprecedented shifts, marine communities face an increasing array of challenges. In this critical context, delving into the investigation and identification of tolerance thresholds for environmental factors becomes mandatory for safeguarding the balance of natural ecosystems. Particularly, with regards to temperature, understanding and pinpointing the key tolerance limits of species and communities is instrumental for comprehending the potential responses of marine ecosystems to both present and predicted future changes, since it may shed light on the thresholds beyond which organisms/communities experience a notable decline in their functioning. Such thresholds, often referred to as “critical thermal limits”, serve as crucial indicators of the resilience or vulnerability of ecological systems in the face of environmental perturbations.

The effects of elevated temperature on sponges’ performance are thought to be species-specific [30]. For example, the sponges *Cliona celata* and *Mycale grandis* were reported to be resistant to temperatures of up to 4–5 °C higher than ambient temperatures [31,32]. There is also evidence that some sponges can even become dominant in some marine tropical environments as a result of extreme thermal events [33,34]. In the Caribbean Sea, for example, sponges belonging to the genus *Chondrilla* became the dominant species in the reef, after thermal anomalies due to the 1998 El Niño event had caused severe bleaching and mass mortality of corals [35]. On the contrary, other species present relatively low thermal tolerance, such as the Mediterranean sponges *Crambe crambe* and *Petrosia ficiformis* whose upper thermal limit was determined to be 26 °C [36]. Although sponges are proposed as putative winners under climate change scenarios [34,37], many species experience physiological stress when exposed to elevated temperatures [30]. Studies assessing the effects of elevated temperatures on sponges through controlled, manipulative experiments are lacking, and they are mostly limited to tropical species [38,39,40]. The lethal effects of high temperatures include extensive bleaching and disease, loss of symbionts, and increased necrosis and mortality [39,40,41,42]. The sub-lethal effects of elevated temperatures include decreased growth and bioerosion rates [42], increased metabolic rates [39,40], and reduced filtering efficiency and pumping rates [38].

In the Mediterranean Sea, sponges can dominate some shallow rocky areas [43]. Despite the importance of sponges in this region, to date few manipulative studies have been carried out to investigate how elevated temperature might affect the metabolic machinery of this benthic group. Here, we used *Chondrilla nucula* [44] as a model species to explore the role of increasing temperature on one of the most common sponges in the Mediterranean Sea. *Chondrilla nucula* is a photophylous encrusting sponge, commonly known as the carpet sea. This species is widely distributed in coastal waters, living in shallow waters from the surface up to ~30 m depth and forming large encrusting patches on well-lit hard substrata. The organism is filter-feeding, actively pumping water through its body to extract suspended particles, and it plays a crucial role in maintaining water quality by influencing nutrient cycling in marine ecosystems. It is a strong competitor for space, a feature that makes it a dominant species in some benthic habitats [43]. *Chondrilla nucula* is also of high commercial interest for the presence of bioactive compounds, which make it an eligible candidate for bioremediation [45]. Due to its shallow distribution, it is potentially exposed to current temperature spikes, marine heat wave effects, and sea-surface temperature increase under CC scenarios. Thus, we tested the species response (in terms of respiration and clearance rates) to short-term exposure to different increasing/decreasing water temperatures (ranging from 15 to 32 °C) simulating different temperature spike conditions.

## 2. Materials and Methods

### 2.1. Sponge Collection and Preparation

Around 100 specimens of *C. nucula* (fragments of ~4 × 4 cm in size) were collected from 5 to 10 m depth by SCUBA divers from the northern coast of Sicily, Italy. Upon collection, sponges were transported to the Ecology Laboratory at the University of Palermo, Italy, and maintained in aerated aquaria with seawater maintained at the same temperature of the collection site (22 °C). Each aquarium (8 L) contained 4–5 sponge samples. Epibionts were carefully removed from the sponge surfaces so that the successive measures (respiration and clearance rates) would not be compromised. Sponges were then randomly assigned to six temperature treatments: 15, 19, 22, 26, 28, and 32 °C (*n* = 8 for each one, divided into two aquaria for each treatment level). After one day of acclimation in the aquaria at 22 °C, the water temperatures were slowly changed (rate of 0.25 °C per hour based on [46,47,48,49] and unpublished data on study site thermal series) to the experimental temperatures, at which sponges were exposed for further 24 h, simulating short-intense temperature spikes [48,50,51]. At the end of this period, sponges were sampled for respiration rates’ and clearance rates measurements. Sponges were handled underwater during all the stages to avoid stress associated with air exposure.

### 2.2. Respiration Rates

Respiration rates were measured on 8 replicates for each temperature treatment according to a well-tested experimental protocol adopted by several companion experiments in the last decade (e.g., [52,53,54,55]). To be sure that there was not any stressful interference due to manipulation, we tested if sponges were actively pumping with fluorescein dye before placing them into individual respiration chambers. Sponges were individually placed into 500 mL oxygen-saturated respiration chambers with filtered (Whatman GF/C) seawater and acclimated in the dark for 20 min prior to start the Dissolved Oxygen (DO) readings. Water inside the chambers was mixed by magnetic stir bars at the bottom of a compartment on which each sponge was sitting. Respiration chambers were kept in a water bath to maintain constant water temperature. DO was measured with Firesting O2 (PyroScience, Aachen, Germany) oxygen fiberglass sensors (four sensor connections per logger) and continuously recorded for 1.5 h in the dark, to avoid any potential oxygen production by photosynthetic symbionts. Blank incubations (*n* = 2 for each temperature treatment) were carried out to adjust actual respiration measurements for any background microbial respiration in the seawater.

Respiration rate (*RR*, μmol O_2_ h^−1^g^−1^ AFDW] was calculated as
*RR* = (C_t0_ − C_t1_) · Vol_r_ · 60 · (t_1_ − t_0_)^−1^
according to Sarà et al. 2013 [52], where C_t0_ was the oxygen concentration at the beginning of the measurement, C_t1_ was the oxygen concentration at the end of the measurement, Vol_r_ was the volume of water in the respirometric chamber, and 60 refers to the measurement time. Respiration rates were standardized for the sponge ash free dry weight (AFDW) after drying sponge samples at 60 °C until constant weight, followed by ashing at 500 °C for 5 h to obtain the organic weight.

### 2.3. Clearance Rates

Clearance rates were measured on 8 replicates for each temperature treatment, which were not the same as those used for the respiration measures. Sponges were individually placed into one-liter chambers and acclimated for 20 min. Feeding chambers were maintained in a water bath to maintain constant water temperature. Water mixing within the chambers was achieved using magnetic stir bars. After acclimation, *Isocrysis galbana* cells were added to the feeding chambers to obtain a concentration of 2.5 × 10^4^ cells mL^−1^. Water samples (20 mL) were taken immediately after microalgae addition (T_0_) and every 30 min for two hours (T_1_, T_2_, …, T_end_). Control feeding chambers (*n* = 2 for each temperature) were used to correct for any drop in cell concentration. Water samples were analyzed with a Coulter Counter to assess particle concentration. Clearance rates [*CR*, l h^−1^] were calculated, following Coughlan equation [56], using the following formula:*CR* = (ln C_1_ − ln C_2_)/time
where C_1_ and C_2_ are the cells’ concentration at the beginning and at the end of each time interval. Clearance rates were standardized for the sponge AFDW, following the methods described above.

### 2.4. Statistics and Modelling

#### 2.4.1. Statistical Analysis

The effect of temperature on respiration rates was tested with a one-way ANOVA. Respiration rate data were log-transformed to meet the normality of the residuals assumption. Tukey’s *post-hoc* pairwise comparisons were conducted to assess whether significant differences between treatments occurred. Clearance rate data were square-root transformed to meet the normality of the residuals assumption. The homogeneity of variance assumption was not met, even after data transformation; therefore, the effect of temperature on clearance rates was tested with a one-way Welch’s ANOVA [57], which is not sensitive to unequal variance. Games–Howell *post-hoc* pairwise comparisons [58] were conducted to assess where significant differences between treatments occurred. Normality of the residuals and homogeneity of variance were tested with the Shapiro–Wilk and Levene tests [59,60], respectively. Statistical analyses and plots were performed in R version 4.1.1 [61].

#### 2.4.2. Thermal Risk Map

According to obtained results from respiration and clearance rate experiments, it was possible to identify a temperature threshold at which the species presents a performance impairment. Such temperature was used to create a thermal risk map for *C. nucula* at Mediterranean scale. Monthly mean temperature from 2019 to 2022 was downloaded from Copernicus Marine Service Information (https://marine.copernicus.eu/, accessed on 20 November 2023) at high spatial resolution (0.08 × 0.08 decimal degrees) for the whole Mediterranean coastline. A temperature raster layer was created, and the mean annual number of days exceeding the identified thermal threshold for the species was calculated at pixel level. It was categorized into 6 different groups, where 0 (green colored) represented any days in that area with temperatures below the threshold; and then the other groups were 1–10 days, 11–20 days, 20–30 days, 31–40 days, and >40 days (red colored) with temperatures above the thermal threshold. Analysis and map were performed in R version 4.1.1.

## 3. Results

### 3.1. Respiration Rates

Temperature had a significant effect on *C. nucula* respiration rates (*F*_(5,42)_ = 82.76, *p* = <0.001). Sponge respiration rates significantly increased with temperature (Figure 1). The lowest respiration rates were measured in the 15 °C treated sponges, while the highest respiration rates were observed in the 26 °C treatment group (2.73 ± 0.59 vs. 75.1 ± 29.93 µmol O_2_ h^–1^ g (AFDW)^–1^, mean ± 95% CI), revealing a clear and responsive relationship between temperature variations and sponge respiration. Sponge respiration rates under the 15 °C treatment were significantly lower than those of all the other treatments groups; while the respiration rates for the 26, 28 and 32 °C treatments (with non-significant differences among them) were significantly higher than the respiration rates measured in the 19 and 22 °C treatments (Figure 1).

### 3.2. Clearance Rates

Temperature had a significant effect on *C. nucula* clearance rates (*F*_(5,18)_ = 25.235, *p* < 0.001) (Figure 2 and Appendix A). The lowest clearance rates were measured in the 15 °C treatment, whereas the highest clearance rates were measured in the experimental sponges belonging to the 26 °C treatment group (0.13 ± 0.035 vs. 0.5 ± 0.192 L h^–1^ g (AFDW)^–1^, mean ± 95% CI; Figure 2). The clearance rates in the 15 °C treatment were significantly lower than those of the 19 and 22 °C treatments; and decreased in the 28 and 32 °C treatments compared to the values recorded in sponges in the 19−26 °C treatments.

### 3.3. Thermal Risk Map

The lowest respiration rates were observed at 15 °C, aligning with the lowest clearance rates. Conversely, the highest respiration rates were associated with the maximum feeding rates for the 26 °C treatment. The higher respiration rates recorded at 28 °C and 32 °C did not signify elevated metabolic rates linked to feeding activity. Instead, they likely reflected a response to stressful environmental conditions. Based on these findings, a temperature of 28 °C was selected as the thermal threshold for creating a risk map reflecting the number of days (at the pixel level) that the temperature exceeded 28 °C.

The thermal risk map (Figure 3) revealed that over 75% of the Mediterranean coast experiences 20 or more days per year with temperatures exceeding 28 °C. The southern coast of the basin, especially the Tunisian and eastern Mediterranean coast, exhibits the highest incidence of extreme temperatures. Conversely, the southern Aegean Sea, the Croatian coast, the Gulf of Lion, and the Alboran Sea maintain annual temperatures below this threshold. The remaining coastline, with a particular emphasis on northern Sicily, registers between 15 and more than 40 days of elevated temperatures.

## 4. Discussions

Using controlled laboratory experiments, we investigated, for the first time, the responses of the Mediterranean shallow-water sponge *Chondrilla nucula* to a range of temperatures, including acute stressful thermal conditions due to marine heat spikes projected under climate change scenarios. Respiration rates significantly decreased at the lowest tested temperature (15 °C), while they increased at higher temperatures, even if they did not significantly vary among the 26, 28, and 32 °C treatments (being the last above normal seawater temperature, 4 °C higher than the maximum seawater temperature recorded in summer 2009). On the contrary, clearance rates decreased at the two highest temperatures (28 °C and 32 °C), indicating possible negative consequences for the energy balance of this species at temperatures equal to or higher than 28 °C. No signs of necrosis were observed in the experimental organisms during the experiment.

Higher respiration rates in sponges were reported in summer seasons, from in situ measurements, both in tropical and temperate regions [62,63], showing that temperature exerts strong control over sponge metabolic rates. Different manipulative experiments focused on long and short-term exposure to thermal stressors observed equally increasing sponge respiration rates at the highest temperature treatments (most of them mirroring ocean warming conditions). Beepat et al. 2020 [39] reported increased respiration rates in the sponges *Neopetrosia exigua* between 26 °C (control temperature) and 30 °C, and in *Amphimedon navalis* and *Spheciospongia vagabunda* between 26 and 28 °C, after two weeks of thermal stress exposure. Similarly, Bennett et al. 2017 [40] reported significantly higher respiration rates in the tropical sponges *Carteriospongia foliascens*, *Rhopaloeides odorabile* and *Cymbastela corallophila* at 31.5 and 30 °C compared to the control temperature, 28.5 °C. Beepat et al. 2021 [64] observed higher oxygen consumption rates for three different sponge species after short-term exposure to 26 °C, 28 °C, and 30 °C temperature treatments. Only a few studies investigated the effects of temperature on sponge clearance rates, obtaining contrasting results. Reduced clearance rates in response to elevate temperatures were reported in the tropical sponge *Rhopaloeides odorabile* after exposure to temperatures 3 °C higher than the control, under laboratory manipulation [38]. On the contrary, an increase in clearance rates was reported in the temperate sponge *Halichondria panicea* at 12 °C compared to 6 °C, in laboratory conditions [65]. Finally, an in situ study did not report altered clearance rates in Mediterranean sponges as a function of temperature, under seasonal temperature ranges [66]. While the response of sponge feeding to thermal anomalies is probably species-specific, future increased sea surface temperature is likely to reduce the ability of sponges to actively pump water, jeopardizing their functioning.

A sponge’s ability to feed, as well as respiration, is directly linked to sponge pumping rates [67]. Massaro et al. 2012 [38], who reported a decline in clearance rates in sponges exposed to increasing temperatures, also reported a decrease in pumping rates. Beepat et al. 2021 [64] reported increased pumping rates along with increased respiration rates in tropical sponges exposed to thermal stress. As respiration rates and clearance rates are linked to pumping rates, we expected these two traits should follow similar patterns in response to different temperatures. While, in the present study, the lowest respiration rates corresponded to the lowest clearance rates at 15 °C, and the highest respiration rates corresponded to the highest feeding rates in the 26 °C treatment, this correlation ceased at temperatures > 26 °C. The higher respiration rates at 28 °C and 32 °C did not indicate elevated metabolic rates related to feeding activity, but they were probably a response to stressful environmental conditions.

Respiration rates represent a measure of the part of the food intake required to provide energy to support life processes [29]. In our study, the contrasting response of two related functional traits may indicate a greater energy expense than the energy gained through food as an immediate response to a short-duration thermal stressor. Such a response may indicate the proximity of the upper thermal tolerance limit for the species, triggering an energy imbalance under stressful conditions. The effects under recurrent and chronic stressor exposure could be more severe, as less energy might be directed toward processes related to an organism’s life-history traits, such as growth and reproduction [68], dealing with consequences at the population level.

Mechanisms to adapt to thermal stress were studied in different sponges’ species, confirming a certain ability to recover after stressful conditions [69]. Even if a larger degree of stress may cause the metabolic defense/compensation system to collapse, some sponge species showed their survival ability by reducing cellular activity under intense thermal stress [70], indicating some adaptability to recover from significant heat stressful conditions [71]. Nevertheless, in the current and future scenarios of CC, temperature spikes and heat waves are predicted to increase in frequency and intensity [9], influencing species responses, acclimation, and adaptation mechanisms. The scientific evidence supporting massive mortality events in marine benthic environments due to stressful thermal events underscores the vulnerability of marine ecosystems to rising sea temperatures. Numerous studies documented the detrimental impacts of elevated sea surface temperatures on marine benthic communities [8,72,73]. Such studies highlighted the complexity of sponge responses to thermal stress, with some species exhibiting bleaching and necrotic tissue, while others appear resilient and unaffected [40,74,75]. Particularly, sponge species in association with phototrophic symbionts (such as *C. nucula*) may present increasing respiration rates and decreasing photosynthetic rates when exposed to high temperatures, even if, in some cases, ocean acidification may mitigate temperature stress in phototrophic species [17,40]. This variability underscores the importance of understanding species-specific reactions to thermal stress and the potential consequences for broader ecosystem dynamics. Species inhabiting environments near their upper thermal limits may face challenges in resisting or compensating for prolonged thermal stress, particularly when compounded by other local stressors. The cumulative effects of these stressors can lead to significant impacts at both the population and community levels, potentially resulting in mass mortality events that disrupt ecosystem functioning.

The outputs from these experiments may indicate potential negative consequences on the energy balance of *C. nucula* at temperatures equal to or higher than 28 °C. As was shown by the thermal risk map, these thermal conditions have been repeatedly verified in recent years within the Mediterranean basin and may suppose a consistent threat as the frequency and intensity of extreme thermal events increase [9]. Even if we present the species response to short-term thermal stressful conditions here, temperature conditions related to future scenarios of climate change may jeopardize the species’ occurrence, abundance, and functioning, severely impacting local populations. Given the high density of the species in some areas of the Mediterranean Sea, the community structure, the associated local biodiversity, and the functioning of the entire ecosystem could be at risk. As global temperatures continue to rise, and extreme thermal events increase in number and frequency, understanding and predicting the responses and tolerance range of marine organisms to temperature becomes paramount. The Mediterranean Sea is experiencing an acceleration of climate change impacts, and the use of mechanistic trait-based approaches to investigate species or communities’ responses to environmental changing conditions may represent the most reliable and best ecologically informed approach in producing realistic predictions about the fate of biodiversity in the Mediterranean Sea. It is mandatory to highlight the urgency of considering the broader ecological implications of species’ responses, emphasizing the need for comprehensive management strategies that account for both local and global stressors, and underscoring the importance of proactive conservation efforts to mitigate the potential cascading effects of thermal stress on marine ecosystems and the valuable services they provide.

## 5. Conclusions

This study demonstrates that the Mediterranean shallow-water sponge *Chondrilla nucula* has a rapid response to change in temperature, with altered respiration and clearance rates. Respiration rates increase with temperature, while clearance rates significantly decrease at temperatures higher than 26 °C. The different patterns of respiration and clearance rates suggest greater energy expenses than the energy gained through food intake at higher temperatures in response to an acute short-term thermal stressor. Although sponges were proposed as putative winners under climate change scenarios, in the Mediterranean Sea, even short-duration temperature anomalies may have detrimental effects on sponges’ communities and overall benthic biodiversity. To gain a more comprehensive understanding of the species’ adaptive capacities, further research must delve into the thermal tolerance thresholds of *Chondrilla nucula*. This exploration is imperative to unravel its potential responses to chronic or repeated exposure to thermal stress, circumstances that could exert a sustained negative impact on the populations of this habitat-forming species and, consequently, on the broader ecosystem’s functioning. As climate change continues to reshape environmental conditions, these insights into the thermal dynamics of key species contribute valuable knowledge for formulating effective conservation and management strategies in the face of ecological challenges.

## Figures and Tables

**Figure 1 biology-13-00009-f001:**
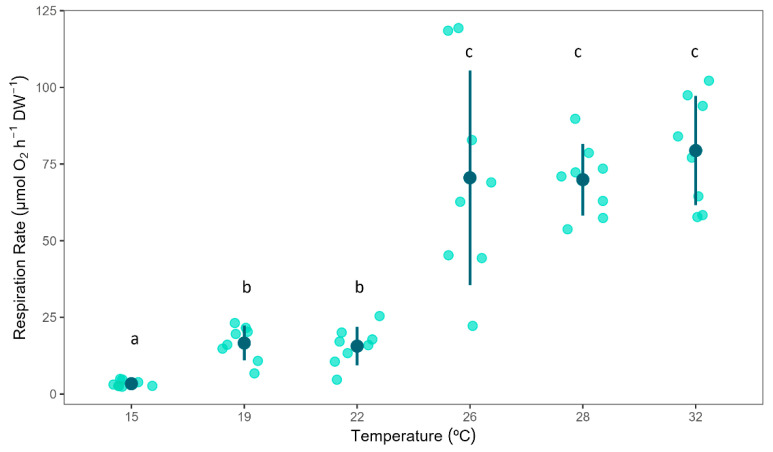
Respiration rates of *Chondrilla nucula* measured at six different temperatures. Bars show mean values ± 95% CI. *n* = 8. Letters indicate significant differences among temperature treatments.

**Figure 2 biology-13-00009-f002:**
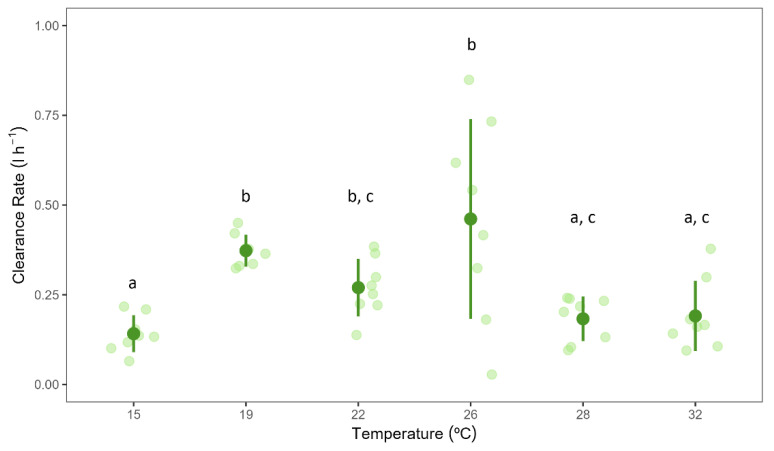
Clearance rates of *Chondrilla nucula* measured at six different temperatures. Bars show mean values ± 95% CI. *n* = 8. Letters indicate significant differences among temperature treatments.

**Figure 3 biology-13-00009-f003:**
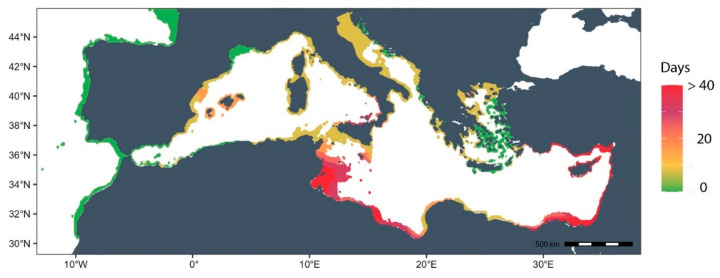
Risk map showing the number of days per year when temperature exceeds the temperature threshold of 28 °C in the Mediterranean Sea. Scale ranges from 0 to 1, where 0 represents any day of temperature above 28 °C and 1 represents more than 40 days exceeding that temperature.

## Data Availability

Data are available in the Mendeley Data online repository (https://data.mendeley.com/datasets/pf75ngh7h7/1, accessed on 15 November 2023).

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
