# Peer review of "Effect of Acute Thermal Stress Exposure on Ecophysiological Traits of the Mediterranean Sponge *Chondrilla nucula*: Implications for Climate Change"

_biology, 2023, doi:10.3390/biology13010009_

Round 1
Reviewer 1 Report (Previous Reviewer 2)
Comments and Suggestions for Authors
As a reply to a reply, in my opinion authors wasted a good explanation in my reply instead of including it into the body of the paper.
The rest of the paper is fine and I see no reason for not publising it as is along with the reviews.
Author Response
We thank the reviewer for the reply and the time dedicated to our manuscript. Information reported in last revision was accordingly included in the main text of the manuscript to clarify and address the issues that the referee highlighted.
Reviewer 2 Report (Previous Reviewer 3)
Comments and Suggestions for Authors
I thank the Authors for addressing the comments to the first version of the manuscript.
The Authors have provided most of the missing information in the methodology. Regarding thermal risk map: please add link and reference to Copernicus Marine Service Information. In L174, I think there is an error and instead of “temperatures exceeding the threshold” the Authors mean “temperatures below the threshold”. Please specify at what depth do these temperature values correspond to.
In the results, figure 1 and 2 have been modified as requested, so that the reader can see the variability in the data. However, note that the new figures are not showing under which conditions there were significant differences. Please add this information, as indicated in the figure legend.
I also appreciate the figure added to supplemental material.
Please upload the raw data to a public repository. The Authors can establish an embargo so that the data will not be publicly available until the manuscript is also published, so there is no need to wait for manuscript acceptance for doing this step.
Below I provided some comments regarding the discussion:
LL211-213: please explicitly state that this experiment evaluates a short-time/peak of temperature stress. In this paragraph, also state if signs of necrosis were observed during the course of the experiment and in which treatment(s).
LL224: what does “CC projection” mean? Does the study refer to specific IPCC projections? If so, cite the corresponding IPCC report
Author Response
We thank the reviewer for the work done and the time dedicated to our manuscript. All suggestions have been addressed, fixing the issues in the main text and including significance within the figures. The data will be soon available in the Mendeley Data repository (Bosch Belmar, Mar (2023), “Chondrilla nucula respiration and clearance rates to changing temperature”, Mendeley Data, V1, doi: 10.17632/pf75ngh7h7.1).
Reviewer 3 Report (New Reviewer)
Comments and Suggestions for Authors
The present manuscript is satisfactory and accepted for publication after minor revisions as follows,
In methodology portion, add the pictures of scuba diving samples collection if you have any, or add the treatment aquarium pictures which you have conducted in the laboratory with sponges samples.
In results portion, please add the descriptive statistical data in table about your 8 aquarium with parameters in range, mean, standard deviation of respiratory rates, clearance rate, and temperature rates. Also found the linear regression relationship between temperature versus respiratory rates, temperature versus clearance rate with statistical significance at p 0.005.
Your work is highly appreciated and suitable for adding new information regarding to sponges biodiversity and their conservation status at Mediterranean Sea coastal area.
Best of luck
Reviewer
Author Response
We thank the reviewer for the reply and the time dedicated to our manuscript. Unfortunately we do not have photographs from the experiments. Information on significance (particularly the post-hoc test) has been included in the ms figures, and all data will be soon available in the Mendeley Data online repository.
This manuscript is a resubmission of an earlier submission. The following is a list of the peer review reports and author responses from that submission.
Round 1
Reviewer 1 Report
Comments and Suggestions for Authors
The authors present data on the effects of temperature on respiration and clearance rates in the sponge Chondrilla nucula, a common Mediterranean sponge. While the topic is interesting, I have concerns about the study design, analysis, and interpretation that preclude recommending publication at this time:
The acclimation period of 24 hours before measurements is inadequate and does not reflect ecologically relevant timescales. A longer acclimation period of 1-2 weeks is standard for sponge thermal studies.
On line 101, the authors write: 'the water temperatures were slowly changed to the experimental temperatures'. However, from the short acclimation (24 hours) and the short duration of the treatment (24 hours), it does not appear that the temperature was raised slowly enough (0.1-0.5 °C per day, so at least 16 days to reach 32 °C from 24 °C). The probably rapid increase in temperature is in itself a stress factor.
The lack of replication at the treatment level is another major issue - each temperature had only one tank, so tank effects are confounded with temperature treatment effects.
The study explores only short-term exposures, yet conclusions extrapolate to climate change scenarios. Long-term acclimatisation potential was not captured, so the study gives little indication of the actual response of C. nucula to ocean warming.
Overall, this study is limited in ecological relevance and methodological rigor. Addressing the above concerns, particularly through a revised experimental design and tempering the conclusions, could make this manuscript more suitable for publication. With its current flaws, I cannot recommend publication.
Author Response
RESPONSES TO REVIEWERS COMMENTS
REVIEWER 1
The authors present data on the effects of temperature on respiration and clearance rates in the sponge Chondrilla nucula, a common Mediterranean sponge. While the topic is interesting, I have concerns about the study design, analysis, and interpretation that preclude recommending publication at this time:
The acclimation period of 24 hours before measurements is inadequate and does not reflect ecologically relevant timescales. A longer acclimation period of 1-2 weeks is standard for sponge thermal studies.
The 24 h “acclimation period” before measurements referred to the temperature treatment, the simulation of marine heat spikes. Treatment duration was at that time decided based on old literature and on own scientific experience developed in more than 20 years of working with sponges. Nevertheless, the current literature supports this choice as for example Hobday et al. 2016 define “temperature spike” (short duration temperature spikes less than five days) and such an approach has fed dozens of scientific articles published in the last decade such as Juza et al. 2022 and Simon et al. 2022 who observed high frequency of marine heat spikes and cold spells over the year in the Mediterranean Sea and Alsuwaiyan et al. 2021 that experimentally tested the effect of this thermal stressor. We have noticed that in this case the term “acclimation” can be confusing, so it has been changed in the main text and more information regarding the treatment duration has been added in the methods section (L136-142).
On line 101, the authors write: 'the water temperatures were slowly changed to the experimental temperatures'. However, from the short acclimation (24 hours) and the short duration of the treatment (24 hours), it does not appear that the temperature was raised slowly enough (0.1-0.5 °C per day, so at least 16 days to reach 32 °C from 24 °C). The probably rapid increase in temperature is in itself a stress factor.
The idea was to test the immediately response of sponges to short-term thermal stressor (realistic conditions in a context of climate change). Temperature was changed at a rate of 0.25 ºC/h based on observed real temperature changing rates when this phenomenon takes place. Moreover, the selected temperature change rate can be considered conservative with respect to already published literature where temperature variations range from 0.33 ºC/h to 2ºC/h (Prusina et al., 2014; Montalto et al., 2017, Alsuwaiyan et al. 2021, Bosch-Belmar et al. 2022). It has been clarified in L136-140.
The lack of replication at the treatment level is another major issue - each temperature had only one tank, so tank effects are confounded with temperature treatment effects.
Thank you for pointing this out. In fact, it was not specified in the manuscript that the acclimation and treatment time for each temperature were done in two aquariums for each treatment level. We have now specified this in the text (L136). Moreover, temperature was maintained in each aquarium through highly precise heater instruments (Lauda A100) and was monitored by temperature data loggers (with minute as recording time frequency).
The study explores only short-term exposures, yet conclusions extrapolate to climate change scenarios. Long-term acclimatisation potential was not captured, so the study gives little indication of the actual response of C. nucula to ocean warming.
We are aware on the limitations regarding the short-term exposure treatment. In fact, in the discussion and conclusions sections we highlighted that the outputs referred to the immediate response of the sponge to environmental temperature changes that may occur due to short-term temperature spikes. Observing an immediate potentially negative effect on the species performance after short exposure time has allowed us to hypothesize that in a future scenario where frequency and intensity of such extreme temperature events will increase, the ability of the species to respond from a functional trait’s point of view will be affected. Moreover, the performed experimental approach allowed us to identify a potential upper thermal threshold of this species (i.e. 28 ºC), that once exceeded may lead to species performance impairment. Such climatic condition is already verified in a large part of the Mediterranean (Fig. 3 of the ms) basin that presents a mean of more than 20 days/year with temperatures higher than 28ºC.
Reviewer 2 Report
Comments and Suggestions for Authors
I have read the "Effect of acute thermal stress exposure on functional traits of the Mediterranean sponge Chondrilla nucula: implications for climate change" manuscript and I can say that is valuable and deserves publication.
The authors study the physical stress on Chondrilla nucula to different temperatures for a day when respiration rate and clearance rate were recorded. The authors extracted valuable pieces of information from their study.
The manuscript was of particular interest for me since I visited recently the metropolitan area covered of this study.
I have only a series of minor revisions suggestions, from which a series of them are of style.
- please check everywhere for numbers in units to have proper representation (changes required in l. 62, l. 88, l. 165, etc).
- use superscript and subscript properly for O_{2}, for h^{-1}, etc. (l. 122)
- use the same style everywhere in eqs (l. 123)
- explain the quantities involved in eqs (l. 123); use an explicit sign for multiplication when all the formulas are simple enough (l. 123)
- revise eq. in line 142 and use square brackets for units instead of round ones; also place the unit at the end of the formula; move the unit for the variable "time" in explanations.
- in all eqs all variables (in latin symbols) should be in italic
- log_{e} is not an usual notation; use "ln" instead
- in "2.4. Statistical analysis" please use the same good behavior to provide references when citing procedures.
- line 161: F, p: in italic; " = < " -> " <" (or " = " providing its approximate value).
"2.73 ± 0.3" and the following - since you used Welch test please replace SE with confidence interval at a certain (let's say 95%) probability.
also in the plots please use confidence intervals - it is much more illustrative, overlapping of the confidence interval meaning no significant difference detected.
Use numbered references style.
There are some more powerful statistical tests available. For instance when comparing two samples you may wish to test if the two samples belong to the same population or not (e.g. if it may come from same sampling conditions). k-Sample Anderson-Darling test will do the trick. You should cover this issue at least in discussions. See supplementary material of Mathematics 2020, 8(2), 216 located at https://www.mdpi.com/2227-7390/8/2/216/s1. Please mention it in your study.
Comments on the Quality of English Language
Are some style errors which must be corrected.
Author Response
RESPONSES TO REVIEWERS COMMENTS
REVIEWER 2
I have read the "Effect of acute thermal stress exposure on functional traits of the Mediterranean sponge Chondrilla nucula: implications for climate change" manuscript and I can say that is valuable and deserves publication.
The authors study the physical stress on Chondrilla nucula to different temperatures for a day when respiration rate and clearance rate were recorded. The authors extracted valuable pieces of information from their study.
The manuscript was of particular interest for me since I visited recently the metropolitan area covered of this study.
I have only a series of minor revisions suggestions, from which a series of them are of style.
- please check everywhere for numbers in units to have proper representation (changes required in l. 62, l. 88, l. 165, etc).
Thanks for the observation, the number units have been revised and standardized throughout the manuscript.
- use superscript and subscript properly for O_{2}, for h^{-1}, etc. (l. 122).
Accordingly modified.
- use the same style everywhere in eqs (l. 123).
Accordingly modified.
- explain the quantities involved in eqs (l. 123); use an explicit sign for multiplication when all the formulas are simple enough (l. 123).
Accordingly modified.
- revise eq. in line 142 and use square brackets for units instead of round ones; also place the unit at the end of the formula; move the unit for the variable "time" in explanations.
Accordingly modified.
- in all eqs all variables (in latin symbols) should be in italic.
Accordingly modified.
- log_{e} is not an usual notation; use "ln" instead.
Accordingly modified.
- in "2.4. Statistical analysis" please use the same good behavior to provide references when citing procedures.
Accordingly modified.
- line 161: F, p: in italic; " = < " -> " <" (or " = " providing its approximate value).
Accordingly modified.
"2.73 ± 0.3" and the following - since you used Welch test please replace SE with confidence interval at a certain (let's say 95%) probability.
Accordingly modified.
also in the plots please use confidence intervals - it is much more illustrative, overlapping of the confidence interval meaning no significant difference detected.
Accordingly modified.
Use numbered references style.
Accordingly modified.
There are some more powerful statistical tests available. For instance when comparing two samples you may wish to test if the two samples belong to the same population or not (e.g. if it may come from same sampling conditions). k-Sample Anderson-Darling test will do the trick. You should cover this issue at least in discussions. See supplementary material of Mathematics 2020, 8(2), 216 located at https://www.mdpi.com/2227-7390/8/2/216/s1. Please mention it in your study.
We appreciate the reviewer’s suggestion on statistical analysis but we prefer to do not change the tests we used. In the case of the analyses conducted on respiration and clearance rates, we decided to use an ANOVA and a nonparametric test such as Welch's, instead of the suggested test because the k-sample Anderson-Darling analysis assesses whether shapes of two distributions are different, but we are interested in whether the mean value of the two distributions differ significantly.
Reviewer 3 Report
Comments and Suggestions for Authors
This work assesses the effect of elevated temperature on sponge filter feeding (as clearance rates) and respiration rates on the Mediterranean sponge Chondrilla nucula. These physiological traits were quantified at 6 different temperatures in controlled experimental conditions in aquaria. Clearance rate were measured as removal of protists over a 2h period. Respiration rates were quantified in individual chambers. Results suggest that respiration rates dropped at temperatures equal or above 26°C, whereas clearance rates apparently increased with temperature until 26°C but then dropped to levels similar to the lowest temperature tested.
The introduction is well written and provide appropriate context to the study. However, different aspects of the methodology need clarification and I have specific requests regarding the presentation of results. The discussion reads nicely and it is concise but compiling.
Comments to material and methods
1. In the current version (L102), it is stated that conditions were “slowly changed to the target temperature”. However, this statement is too vague. Please provide specific detail on how temperature was increased to reach the target temperature in each treatment (e.g.: as X°C per hour or °C per day). Also, provide specific detail on how the temperature was increased and decreased to reach the targets (e.g., brand of heaters/coolers) and how were temperature conditions monitored over time.
2. In line 93 it is stated that 50 sponge specimens were collected from natural populations. And later, that there were 8 replicate per treatment for each of the two physiological measures: “Clearance rates were measured on 8 replicates for each temperature treatment which were not the same of those used for the respiration measures.” (Lines 131-132). This experimental design requires at least 2x (6 temperature conditions x 8 biological replicates) = 96 specimens. Please correct/clarify this apparent contradiction. Also, it is unclear to me if whole individuals or rather sponge fractions were collected from the field.
3. What type of seawater was used for the experiments (e.g., filtered/unfiltered natural seawater, artificial seawater)?
Comments to presentation of results:
4. It is advisable to replace the bar charts with plots that show the real distribution of Y values for each temperature. This can be as a dot +box plot, or dot plot with means ± SE as whiskers, or a violin plot showing also the means.
5. Please add a plot showing the cell concentrations over time for each sponge replicate and for the controls. This would be a figure 2, and clearance rate plot would be figure 3.
6. Raw data should be deposited in a public repository (e.g., dryad, figshare) or as an excel/table file in supplemental material.
7. It would be interesting to report at the beginning of the result section how the sponges look like at the different temperature treatments. Did you observe any signs of necrosis? Maybe consider adding some pictures either to the main manuscript or as supplemental material.
Comments to discussion:
8. In lines 222-229, the Authors discuss the correlation between respiration and clearance rates and I wonder if it may be helpful to add a plot to show the different correlation trend below and above 26°C. This pattern may be interesting to readers and in may encourage testing it in different sponges both in situ and ex situ.
9. Lines 230-238: how it is this statement be affected if we take into account that this sponge species is harboring photosymbionts?
Suggestion:
This is just a thought. I wonder if in the title and throughout the manuscript “functional traits” should be replaced by “physiological traits”. This may be perceived different depending on the background of the reader, but I feel “functional traits” in sponge research would refer to nutrient cycling capabilities or production of secondary metabolites or microbial functions, whereas respiration rates and clearance rates are referred as physiological measurements.
Author Response
RESPONSES TO REVIEWERS COMMENTS
REVIEWER 3
This work assesses the effect of elevated temperature on sponge filter feeding (as clearance rates) and respiration rates on the Mediterranean sponge Chondrilla nucula. These physiological traits were quantified at 6 different temperatures in controlled experimental conditions in aquaria. Clearance rate were measured as removal of protists over a 2h period. Respiration rates were quantified in individual chambers. Results suggest that respiration rates dropped at temperatures equal or above 26°C, whereas clearance rates apparently increased with temperature until 26°C but then dropped to levels similar to the lowest temperature tested.
The introduction is well written and provide appropriate context to the study. However, different aspects of the methodology need clarification and I have specific requests regarding the presentation of results. The discussion reads nicely and it is concise but compiling.
Comments to material and methods
- In the current version (L102), it is stated that conditions were “slowly changed to the target temperature”. However, this statement is too vague. Please provide specific detail on how temperature was increased to reach the target temperature in each treatment (e.g.: as X°C per hour or °C per day). Also, provide specific detail on how the temperature was increased and decreased to reach the targets (e.g., brand of heaters/coolers) and how were temperature conditions monitored over time.
All requested information have been included in the text from L136 to L142. - In line 93 it is stated that 50 sponge specimens were collected from natural populations. And later, that there were 8 replicate per treatment for each of the two physiological measures: “Clearance rates were measured on 8 replicates for each temperature treatment which were not the same of those used for the respiration measures.” (Lines 131-132). This experimental design requires at least 2x (6 temperature conditions x 8 biological replicates) = 96 specimens. Please correct/clarify this apparent contradiction. Also, it is unclear to me if whole individuals or rather sponge fractions were collected from the field.
Thanks for pointing out the issue, as the reviewer highlights 50 sponge fractions were used for each experimental set, then a total of 100 experimental units were sampled. It has been correct in the main text (L129). - What type of seawater was used for the experiments (e.g., filtered/unfiltered natural seawater, artificial seawater)?
It was used filtered (Whatman GF/C) seawater (add in L150).
Comments to presentation of results:
- It is advisable to replace the bar charts with plots that show the real distribution of Y values for each temperature. This can be as a dot +box plot, or dot plot with means ± SE as whiskers, or a violin plot showing also the means.
Accordingly modified. - Please add a plot showing the cell concentrations over time for each sponge replicate and for the controls. This would be a figure 2, and clearance rate plot would be figure 3.
Thanks for the suggestion, we have included this figure as supplementary material. - Raw data should be deposited in a public repository (e.g., dryad, figshare) or as an excel/table file in supplemental material.
Thanks for the suggestion, we already considered the issue and raw data will be upload to Dryad online repository if manuscript will be accepted for publication. - It would be interesting to report at the beginning of the result section how the sponges look like at the different temperature treatments. Did you observe any signs of necrosis? Maybe consider adding some pictures either to the main manuscript or as supplemental material.
Thanks for such informative suggestion but unfortunately we do not have pictures of experimental sponges under temperature treatments.
Comments to discussion:
- In lines 222-229, the Authors discuss the correlation between respiration and clearance rates and I wonder if it may be helpful to add a plot to show the different correlation trend below and above 26°C. This pattern may be interesting to readers and in may encourage testing it in different sponges both in situ and ex situ.
We thank the reviewer for the kind suggestion. We added a graphical abstract to the submission where in an easy and visual way the readers can rapidly observed the opposite trend that clearance and respiration rates showed after 26 ºC. - Lines 230-238: how it is this statement be affected if we take into account that this sponge species is harboring photosymbionts?
Marine sponges respond to increasing temperatures in different ways. Some authors have observed non-effect of warming stressor on sponges physiology and microbiota (Strand et al. 2017, González-Aravena et al. 2019), while other report tissue necrosis and bleaching with changes and/or disruption on the associated photosymbionts (Bennet et al. 2017). In the specific study case of C. nucula we did not observed bleaching or necrotic tissue over the experiment, probably due to the short-term treatment exposure but this cannot be excluded if the stressor properties change. We have include some lines in the text regarding it (L302-309).
Suggestion:
This is just a thought. I wonder if in the title and throughout the manuscript “functional traits” should be replaced by “physiological traits”. This may be perceived different depending on the background of the reader, but I feel “functional traits” in sponge research would refer to nutrient cycling capabilities or production of secondary metabolites or microbial functions, whereas respiration rates and clearance rates are referred as physiological measurements.
Thanks again for the suggestion. We agree with the reviewer and the title has been modified.